

# Incorporating nonlinearity with generalized functional responses to simulate multiple predator effects

Michael W. McCoy[1], Elizabeth Hamman[2], Molly Albecker[3], Jeremy Wojdak[4], James R. Vonesh[5] and Benjamin M. Bolker[6]

[1] Florida Atlantic University (Harbor Branch Campus), Ft. Pierce, FL, United States
[2] St. Mary's College of Maryland, St Mary's City, MD, United States
[3] Utah State University, Logan, United States
[4] Radford University, Radford, United States
[5] Virginia Commonwealth University, Richmond, VA, United States
[6] McMaster University, Hamilton, Canada

## ABSTRACT

Predicting the combined effects of predators on shared prey has long been a focus of community ecology, yet quantitative predictions often fail. Failure to account for nonlinearity is one reason for this. Moreover, prey depletion in multiple predator effects (MPE) studies generates biased predictions in applications of common experimental and quantitative frameworks. Here, we explore additional sources of bias stemming from nonlinearities in prey predation risk. We show that in order to avoid bias, predictions about the combined effects of independent predators must account for nonlinear size-dependent risk for prey as well as changes in prey risk driven by nonlinear predator functional responses and depletion. Historical failure to account for biases introduced by well-known nonlinear processes that affect predation risk suggest that we may need to reevaluate the general conclusions that have been drawn about the ubiquity of emergent MPEs over the past three decades.

## INTRODUCTION

Predicting how changes in predator abundance and diversity influence the structure and function of food webs has remained an elusive target for ecologists for decades (*Ives, Cardinale & Snyder, 2005*; *McCoy, Stier & Osenberg, 2012*; *Sih, Englund & Wooster, 1998*; *Soluk & Collins, 1988*). This difficulty stems in part from the myriad pathways through which predator-prey interactions can unfold. While predators provide essential ecosystem functions *via* prey consumption, they also affect food web dynamics *via* a variety of non-consumptive effects (*Byrnes & Stachowicz, 2009*; *Cardinale et al., 2012*; *Preisser, Orrock & Schmitz, 2007*; *Werner & Peacor, 2003*). Understanding and predicting the roles of predators in food web dynamics becomes even more complicated when considering multiple predators that share prey (*McCoy, Stier & Osenberg, 2012*; *Sih, Englund & Wooster, 1998*). One predator species may induce changes in prey behavior, morphology, physiology, or development that alter the prey's interactions with other predators (*Preisser,*

Corresponding author
Michael W. McCoy,
mccoym@fau.edu

*Orrock & Schmitz, 2007*; *Werner & Peacor, 2003*). Quantifying the emergent effects of multiple predators on prey suppression, defined as deviations from predicted levels of prey suppression based on the independent effects of each predator (*Sih, Englund & Wooster, 1998*), is consequently critical for understanding real ecological communities. However, common approaches for generating null expectations for independent predator effects make many simplifying assumptions that limit the accuracy of our measurement of emergent multi-predator effects (*Sentis & Boukal, 2018*). In this article, we build on recent advances in this field to present a generalized functional response framework that incorporates nonlinear processes that modify prey risk to generate appropriate null expectations for the aggregate effects of multiple predators on shared prey.

Predator consumption of prey is typically described by the predator's functional response, which describes the number of prey eaten in a given period of time as a function of prey density. *Holling (1959)* proposed three general forms for predator functional responses based on the attack rate (*i.e.*, rate that predators encounter and capture prey), and handling time (*i.e.*, the time spent pursuing, consuming, and digesting prey) of the predators. A Type I (linear) functional response assumes that prey consumption by predators is defined by attack rate, whereas Type II (saturating) and Type III (sigmoidal) assume prey consumption is determined by attack rate at low prey densities and handling time at higher prey density (*Holling, 1959*; *Royama, 1971*). Holling's formalization of mechanistic models for predator functional responses revolutionized the study of predation ecology. Since its introduction, the functional response framework has been a mainstay for theoretical and experimental studies in food web ecology, and an important tool for ecological applications such as evaluating the potential effectiveness of biocontrol agents (*Cuthbert et al., 2018*) and impacts of invasive predators (*Dick et al., 2014*). However, the use of Holling's functional responses for quantifying and predicting the strength of predator-prey interactions has had mixed success, due in part to simplifying assumptions that are violated in most natural and experimental settings (*e.g.*, *Griffen, 2021*).

The multiplicative risk model (MRM)—introduced by *Soluk & Collins (1988)* and formalized in a review by *Sih, Englund & Wooster (1998)*—is the most broadly used method to generate null expectations for the combined effects of multiple predators with the MRM approach receiving more than 1,350 citations to date. The MRM predicts the expected proportion of prey surviving in the presence of two independent predators as the product of survival in the presence of each predator alone (*Billick & Case, 1994*; *Griffen, 2006*; *McCoy, Stier & Osenberg, 2012*; *Sih, Englund & Wooster, 1998*; *Soluk & Collins, 1988*; *Vonesh & Osenberg, 2003*). However, appropriate application of the MRM requires either that the predator has a type I (*i.e.*, linear) functional response or, if the functional response is nonlinear, that prey are immediately replaced so that their density is maintained over time and thus their predation risk is constant (*McCoy, Stier & Osenberg, 2012*; *Sih, Englund & Wooster, 1998*).

Most predators have non-linear saturating functional responses (*Jeschke, Kopp & Tollrian, 2004*; *Lafferty et al., 2015*); therefore, in the absence of instantaneous replacement, *per capita* risk for prey changes over time as prey are depleted by predation events (*Juliano,*
*Scheiner & Gurevitch, 2001*; *McCoy, Stier & Osenberg, 2012*; *Rogers, 1972*). Most multiple predator experiments allow depletion of prey, which typically leads to inflated estimates of *per capita* risk due to Jensen's inequality (*McCoy, Stier & Osenberg, 2012*; *Ruel & Ayres, 1999*), and biased predictions of both single and combined predator effects (*McCoy, Stier & Osenberg, 2012*; *Sentis & Boukal, 2018*). In these scenarios, the application of the MRM as a null model likely fails to predict the combined effects of multiple predators correctly, simply because it fails to account for the nonlinearity typical of predation rates across prey abundance (*McCoy, Stier & Osenberg, 2012*). Despite the limitations of the MRM for predicting multiple predator effects identified in *McCoy, Stier & Osenberg (2012)*, the MRM has been cited more than 80 times over the past decade (based on a Google Scholar search of "multiplicative risk model" and predator*).

In addition to prey depletion and a nonlinear functional response, many other nonlinear processes can limit the utility of commonly used approaches for predicting effects of multiple predators on shared prey. For instance, variation in predator and/or prey sizes or variation in environmental temperature can change *per capita* predation risk in nonlinear ways. In the remainder of this article, we expand on approaches presented in *McCoy et al. (2011)* and *McCoy, Stier & Osenberg (2012)* by describing a framework for predicting predator effects on prey based on a generalized functional response that can be used to incorporate a variety of nonlinear processes (*e.g.*, density, size, or temperature dependence) that modify per-capita risk. Next we use simulations to explore how nonlinear risk associated with variation in prey body size may generate biased predictions in multiple predator effects studies that do not account for size dependent predation risk.

## Generalized functional response framework

The generalized functional response framework integrates nonlinear predator functional responses, nonlinear modifiers of risks (*e.g.*, size- or temperature-dependent predation), and changes in somatic growth to generate null expectations for the combined effects of any number of independently acting predators. Specifically, we use a partial differential equation model that allows predation to change as a function of both prey density and a nonlinear modifier ($s$) of predation rates ($E_i$) by $k$ different predator types ($P_i$); and when relevant, it can also account for changes in size-dependent and density-dependent prey somatic growth ($g$).

$$\frac{\partial C(s,t)}{\partial t} = -C \sum_{i=1}^{k} E_i(C,s,P_i) + \left( -g(s)\frac{\partial C}{\partial s} \right) + D\left( \frac{\partial^2 C}{\partial s^2} \right) \tag{1}$$

In this equation, the vector $\boldsymbol{C}$ denotes the distribution of prey abundances across levels of $s$. For simplicity, we assume that predator body sizes within a given species are approximately constant over time.

We assume prey consumption ($E_i$) by each predator ($P_i$) follows a Type II saturating functional response (*Holling, 1959*; *Lafferty et al., 2015*; *McCoy, Stier & Osenberg, 2012*; *Rogers, 1972*), which describes the density-dependent foraging dynamics of most natural enemies (*Jeschke, Kopp & Tollrian, 2004*; *Lafferty et al., 2015*). We incorporate modifier-dependent predation by modeling the attack rate, $a_i$, and handling time, $h_i$, as
functions of a prey risk modifier $s$. Details about the derivation of our size distributed functional response are in Supplement 1.

The total attack rate from predator type $i$ (predators act independently, *i.e.*, there is no interference competition) is

$$E_i = \frac{a_i(s)C(s)}{1 + A_iH_i},$$ (2)

where $A_i = \sum_{s=min}^{max} a_i(s)C(s)$ is the *total* attack rate for predator type $i$ (*i.e.*, the expected per-predator rate of prey consumption when prey are rare or handling is not limiting) and $H_i$ is the propensity-weighted average handling time, *i.e.*,

$$H_i = \frac{\sum_{s=min}^{max} a_i(s)C(s)h_i(s)}{A_i}$$

These expressions take into account the fact that prey-handling by predators is limited by the entire range of prey they consume, not just the prey in a particular class (Supplement 1).

A wide array of potential processes and functional forms can be used to model modifier-dependent predation risk (*e.g.*, *McCoy et al., 2011*). For instance, temperature is known to modify predation in nonlinear ways and could be modeled by making the attack rate a quadratic function of temperature and handling time a power function of temperature and predator body mass (*Davidson et al., 2021*; *Sentis & Boukal, 2018*; *Uiterwaal & DeLong, 2020*).

Here we focus on body size as an exemplar of a modifier that has nonlinear effects on predation risk. Size-specific risk is often a nonlinear function of predator and prey body sizes and can sometimes have stronger implications for per-capita predation risk than prey density (*McCoy et al., 2011*; *McCoy & Bolker, 2008*; *Rudolf, 2008*; *Woodward et al., 2005*). Therefore, failure to account for variation in the sizes of prey in experiments (*e.g.*, differences in prey size structure), or changes in prey size as a result of somatic growth over the course of an experiment (*e.g.*, a study of a single cohort over time), can bias predictions about the combined effects of multiple predators. Most species increase in size by orders of magnitude during their lifetimes (*Fenchel, 1974*; *Werner & Gilliam, 1984*); thus, failure to appreciate the dynamic effects of body size on predator-prey interactions can complicate our understanding of predator prey dynamics.

We will explore two empirically supported functions for changes in predation risk as functions of prey size, an exponential function (*Alford, 1999*; *Aljetlawi, Sparrevik & Leonardsson, 2004*; *McCoy et al., 2011*; *McCoy & Bolker, 2008*) of the form

$$a = \alpha \cdot e^{\left(1-\frac{s}{d}\right)}$$ (3)

and a unimodal function (*Kalinkat et al., 2013*; *McCoy et al., 2011*; *McCoy & Bolker, 2008*; *Vonesh & Bolker, 2005*; *Vucic-Pestic et al., 2010*; *Wahlström et al., 2000*) of the form

$$a = \alpha\left(\frac{s}{d} \cdot e^{\left(1-\left(\frac{s}{d}\right)\right)}\right)^{\gamma}$$ (4)

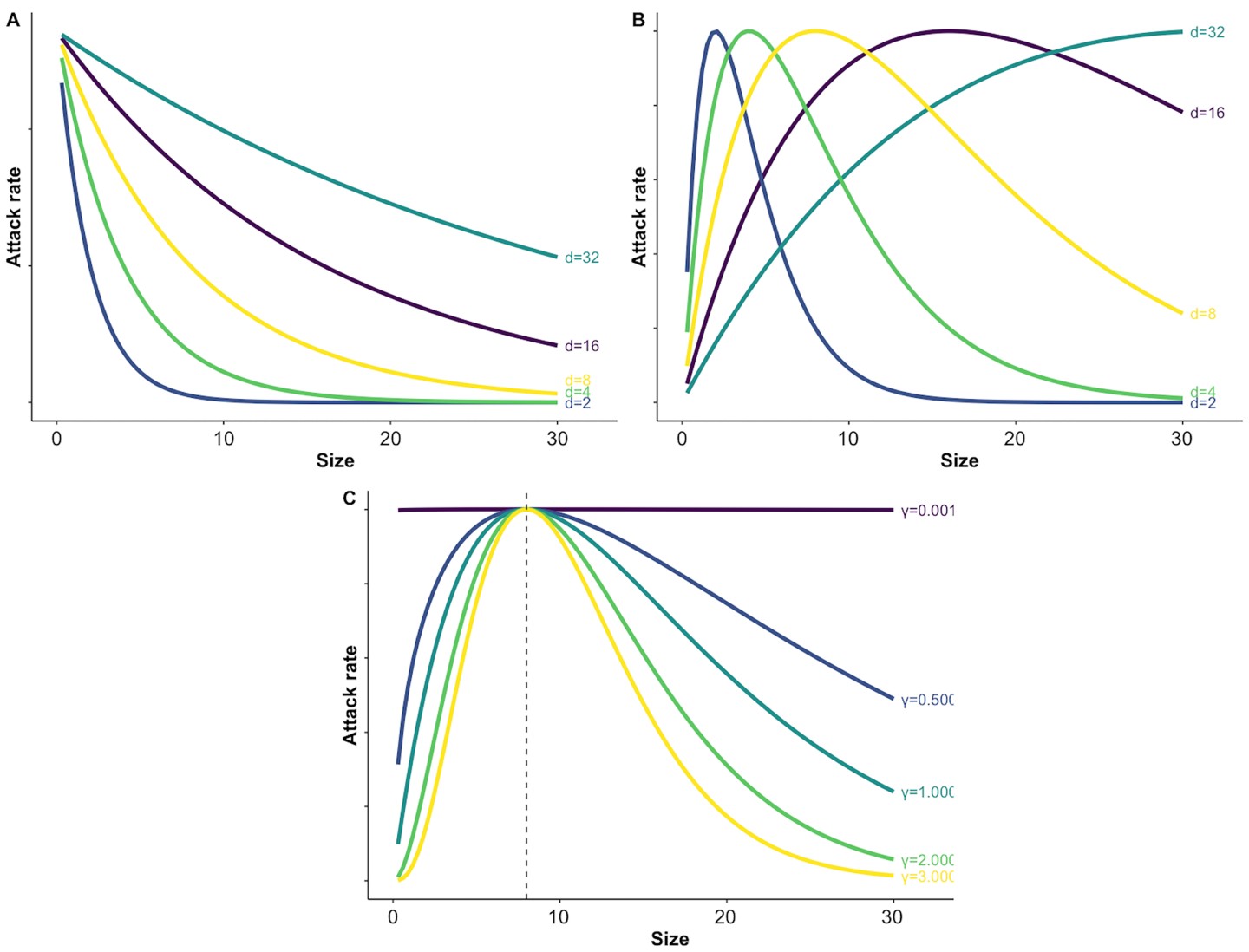

**Figure 1 Illustration of varying parameters of the size dependent risk function and predation risk for prey of different sizes.** Panel A illustrates how changing the value of the size scaling parameter *d* affects the shape of exponential size-dependent risk (Eq. (3)). Panel B demonstrates how changing *d* affects the shape of the unimodal risk function with γ = 1.0 (see C and Eq. (4)). Panel C demonstrates how changing which determines the width of the window of vulnerability affects risk across prey sizes. For C the size scaling parameter *d* was fixed at a value of 8 (see B and Eq. (4)).

where α is maximum attack rate, *d* scales the most vulnerable prey size *s*, and γ determines the width of the window of maximum vulnerability (*i.e.*, the level of prey size specialization for a predator) (Fig. 1). For simplicity, we will assume predator size is not variable, however predator size or predator:prey body size ratios could also be considered within this framework. We assume handling time is independent of prey body size, but this assumption could be relaxed (*e.g.*, Eq. (2) and *McCoy et al., 2011*).

When predation is size-dependent we may also need to consider prey body size growth through time. We represent prey growth in our generalized functional response formulation *via* an advection term that moves the prey through different size classes. In order to integrate these equations over time in a numerically stable way, we implement

the advection term by assuming the locations of size bins (rather than sizes of individuals) change over time. We first solve the growth equation ($ds/dt = g(s)$) to find size as a function of time, then substitute those size values into the predation term at each time step. As with size-dependent predation, many functional forms are available for modeling growth; we use a standard Gompertz growth model. This model assumes that relative growth rate decreases exponentially with size, leading to a growth curve

$$s(t) = \rho \cdot e^{-\theta \cdot e^{(-\delta t)}} \tag{5}$$

where $\rho$ is the maximum size and $\theta$ is the negative log of scaled initial size (*i.e.*, $\theta = -\ln(s(0)/\rho)$) and $\delta$ is a constant that describes the rate of decrease in relative growth rate as size decreases. For example, if the first size bin initially includes individuals in the range [s1(0),s2(0)], at time t it will include individuals in the range [s1(t),s2(t)].

We also incorporate individual variation in growth *via* a diffusion term, the third term in Eq. (1). This term describes diffusion along the size axis that naturally arises from continuously occurring, independent variation around growth rate (*i.e.*, Brownian motion *Brooks, McCoy & Bolker, 2013*). Because the width of the size bins in the model varies over time according to the specified growth curve, the effective diffusion rate is also size-dependent.

For the purposes of this study, we focus on two predator-one prey interactions with a few specific functional forms. However, Eq. (1) is generalizable to model assemblages consisting of many (*i* = 1 to *k* in Eq. (1)) concurrent predators and a wide array of functional forms for the predator functional response (*e.g.*, Holling type III, *Holling, 1959*), risk modifiers, and somatic growth models (*Gompertz, 1825*; *Kahm et al., 2010*; *Richards, 1959*; *West, Brown & Enquist, 2001*).

We conduct a literature review to examine the prevalence of size dependent interactions in multiple predator effects studies. Next, we simulate predation by two predators using our generalized functional response model with prey size-dependent predation risk and compare the simulation results with predictions generated by the multiplicative risk model (MRM). We then explore how different functional forms of size-dependent risk influence the MRM predictions relative to simulation outputs. To contextualize our analysis we used functional relationships and estimates of size variation reported in studies of size dependent predation on red eyed treefrog larvae (*Brooks, McCoy & Bolker, 2013*; *McCoy et al., 2011*).

# METHODS

## Reviewing size dependence

We searched web of science on May 8, 2018 using the terms "("multiple pred\*") OR ("risk reduction" & pred\*) OR ("risk enhancement" & pred\*) OR (MPE & pred\*) & experiment", and retained results for the previous 20 years, a total of 492 papers. We then screened the studies to include only those that were experimental, manipulated presence of at least two predators, and had treatments of control, predator monocultures, and multiple predators foraging together. After screening, we retained 121 observations from 119 studies

(Supplements 2 and 3). We then reviewed the full text of these 121 studies to determine how or if prey size was reported and if so, how much size changed over the duration of the experiment.

## Comparing the MRM predictions and generalized functional response simulations

The MRM predicts the expected proportion of prey surviving in the presence of two independent predators as the product of survival (proportion alive) in the presence of each predator alone ($S_i$ and $S_j$) corrected for survival in the absence of a predator ($S_c$) in additive

$$\grave{S}_{ij} = \frac{S_i \cdot S_j}{S_c} \tag{6}$$

or substitutive

$$\grave{S}_{ij} = \frac{S_i^{0.5} \cdot S_j^{0.5}}{S_c} \tag{7}$$

experimental designs (*Billick & Case, 1994*; *Griffen, 2006*; *McCoy, Stier & Osenberg, 2012*; *Sih, Englund & Wooster, 1998*; *Soluk & Collins, 1988*; *Vonesh & Osenberg, 2003*). Predicted predation rates based on the MRM assume that per-capita risk is the same for all prey individuals and that it remains constant over time (*McCoy, Stier & Osenberg, 2012*; *Sih, Englund & Wooster, 1998*). However, when predators have nonlinear and saturating functional responses (*Jeschke, Kopp & Tollrian, 2004*; *Lafferty et al., 2015*) and prey are depleted, or when risk is a nonlinear function of some other modifying factor (*e.g.*, temperature or size), *per capita* risk is not constant (*Juliano, Scheiner & Gurevitch, 2001*; *McCoy, Stier & Osenberg, 2012*; *Rogers, 1972*). Thus, applications of the MRM based on static estimates of survival can bias estimates of per-capita risk from individual predators, which carry over to bias predictions of combined predators (*McCoy, Stier & Osenberg, 2012*; *Sentis & Boukal, 2018*). By accounting for nonlinear modifiers of prey risk and depletion, we expect that the generalized functional response framework presented here could be used to generate more accurate predictions for the combined effects of multiple predators.

We simulated foraging trials for predators in monoculture and in mixed predator species assemblages by numerically solving Eq. (1) (example code available in Supplement 4), providing estimates of prey survival for each of two predators alone and in combination (*i.e.*, $S_1$, $S_2$, and $S_{1,2}$). Estimates of $S_{1,2}$ from these simulations provided the null model for independent interactions between two predators with size-dependent predation rates. We only simulate additive combinations of predators; thus, all comparisons are based on predictions from the MRM described by Eq. (6) and the generalized functional response in Eq. (1). The degree of mismatch defined as the rate difference between generalized functional response simulations and the predictions from the MRM (Eq. (6)) provides a metric for the potential bias in predictions when prey are being depleted and predation risk is size-dependent. In other words, differences between simulation results and MRM predictions are caused by a failure of the MRM to account for the non-linear, but

fundamentally predictable (and therefore not emergent), effects of density- and size-dependent predation.

For a broad range of plausible parameter values for each predator, we explored how failure to account for size-dependence can bias inferences about multiple predator effects. For attack rates that decayed exponentially with prey size, we varied the size scaling parameter $d$, which describes how risk decreases with prey size (Fig. 1A). For attack rates that were unimodal functions of size, we varied two key parameters regulating size-dependent risk, while keeping all other parameters constant: $d$, the size scaling parameter that determines the most vulnerable prey size (Fig. 1B), and $\gamma$ the parameter that determines the degree of size specialization (*i.e.*, width of the window of maximum vulnerability) (Fig. 1C). Because each of these parameters affects size-dependent risk due to their independent and interactive effects, we explored how they affected the accuracy of predictions in three ways. First, to examine the interactive effects of the unimodal shape parameters, we set one predator to be a size generalist (*i.e.*, a low value $\gamma$; Eq. (4)) that was most effective at killing small prey (*i.e.*, a low value of $d$; Eq. (4)) and then allowed the second predator to have increasingly higher values of these two parameters (*i.e.*, increasingly more size specialized and more effective in consuming larger prey) in each simulation. Second, to explore the independent effects of the values of the size scaling ($d$) and size specialization ($\gamma$) parameters on bias, we fixed both predators to have either high or low values of each parameter (*i.e.*, to be more efficient consumers of large or small prey, $d$; or to be size specialist or generalist, $\gamma$). Third, we allowed the value of the free parameter to vary between the predators.

Given the sparse data on prey size in studies of MPEs, we grounded our simulations by using prey sizes and growth rates published for one particular empirical system (*McCoy et al., 2011*) along with attack rates and handling times to match those used in *McCoy, Stier & Osenberg (2012)* (*i.e.*, maximum attack rate $\alpha = 0.5$, and handling time $h = 1$). We initiated all simulations with a prey population size of 100 individuals evenly divided into 10 prey size classes ranging between 1 and 45 mm. We varied the size scaling parameter $d$ for attack rates that decayed exponentially with size (*via* Eq. (3)). We varied the value of $\gamma$ between 0.001 and 3.5, and the value of $d$ from 3.5 to 24 (see table 2 in *McCoy et al., 2011*). For simulations of growing cohorts we set maximum size $\rho$ (Eq. (5)) at 35 mm, the maximum growth rate parameter $\theta$ to 0.05 and the growth rate decay parameter $\delta$ to a value of $-0.05$ (Eq. (5)), which are within the ranges of estimates of tadpole growth reported in *McCoy et al. (2011)* and *Albecker & McCoy (2019)*.

## RESULTS

Out of 119 studies on emergent multiple predator effects published between 1998 and 2018 (Supplements 2 and 3), only 15% provide information on prey size at all, and only 3% provide numerical descriptions of size ranges of prey. Although studies with significant prey size variation may be more likely to report prey sizes, in those studies reporting data on prey size there was an almost two-fold difference in size on average between the smallest and largest prey (Supplements 2 and 3).

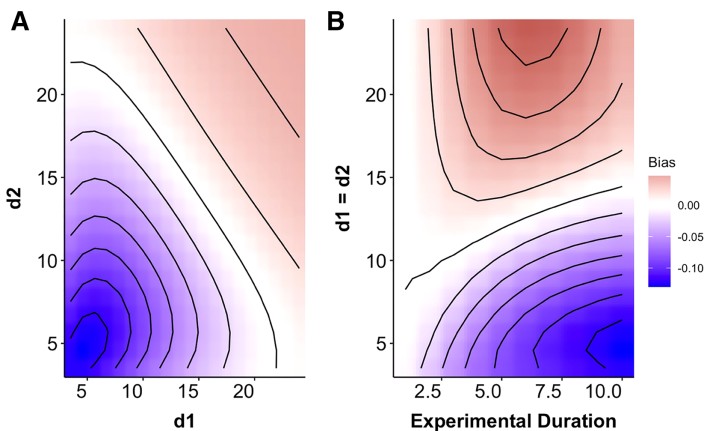

**Figure 2** **Heat map indicating the degree of mismatch between the Multiplicative Risk Model and the generalized functional response.** (A) The x- and y-axes depicts variation in the rate at which predation risk decayed with size for each predator ($d_i$ from Eq. (3)). (B) The magnitude of bias changes over time for two identical predators over a range of exponential decay rates. The patterns in B indicate a progression through time to the conditions depicted along the 1:1 line in A. Negative differences mean that the MRM predicted higher risk to prey than the generalized functional response model (leading to conclusions of risk reduction), whereas positive differences underestimated risk (leading to conclusions of risk enhancement for prey). For instance, after 7 days the MRM would predict 5% higher survival than expected for prey that have weak size dependence and 10% lower survival when prey risk is strongly size dependent.

## Prey size refuge

When predation risk decayed exponentially with prey size, the MRM predicted either higher or lower risk than expected from the generalized functional response model depending upon the specific parameters of the risk function (Fig. 1A) and the experimental duration (Figs. 2A and 2B). For small values of $d$ (*i.e.*, strong size dependence for small prey) the MRM predicted higher levels of risk than expected (*i.e.*, predicted lower survival), but this pattern was reversed for larger values of $d$ (*i.e.*, strong size dependence for larger prey) with the MRM predicting lower risk (higher survival) than expected (Fig. 2). This pattern emerged because the MRM overestimates predation risk for larger prey when risk is strongly size-dependent and fails to capture the higher combined size specific rates of depletion for small and intermediate sized prey when the rate of risk decay is shallower (*i.e.*, larger values of $d$). These patterns were similar for growing cohorts and size structured prey, but with the magnitude of bias for growing cohorts strongly regulated by prey growth rates.

## Size specialization

When size dependence was a unimodal function of prey size and small prey were most vulnerable (*i.e.*, low values for $d$ – Fig. 1B), the MRM predicted lower survival as the predators became more specialized relative to the generalized functional response predictions (Fig. 3A). However, this pattern reversed when either predator was a size generalist, with the MRM predicting higher survival than the generalized model (Fig. 3A). Interestingly, the MRM predicted higher survival over a much broader parameter space as the most vulnerable size class increased, with the pattern only reversing for the most
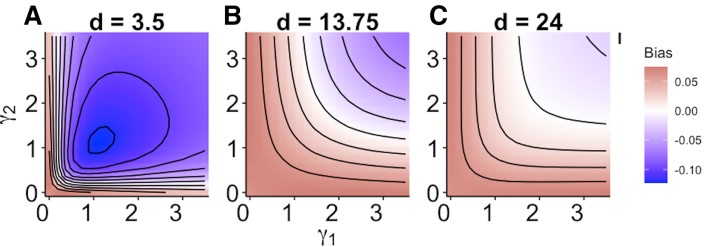

**Figure 3 (A–C) Mismatch in the predictions of the MRM and generalized functional response models when predation risk was a hump shaped function of prey size.** Each facet displays the outcome from simulated experiments that ran for 10 times steps for three different focal prey sizes. The heatmap indicates differences between predicted survival based on the MRM and generalized functional response model. The x and y axes represent the values of the focal size parameter for the two predators Eq. (4). Negative values indicate the MRM predicted higher risk to prey (leading to conclusions of risk reduction), whereas positive values the MRM underestimated risk (leading to conclusions of risk enhancement for prey).            

size-specialized predators (Figs. 3B and 3C). This pattern is likely due to the increasing influence of depletion on *per capita* risk, as a large range of size classes are vulnerable to predation (*McCoy, Stier & Osenberg, 2012*). The strong effects of specializing on small prey also emerge in Fig. 4. When predators do not have strong size preferences (Fig. 4A; small $\gamma$) the MRM predicts higher survival for all values of $d$ because it fails to account for the increasing risk that occurs with depletion (*McCoy, Stier & Osenberg, 2012*). However, this pattern reverses as the predator size specialization becomes more humped shaped and peaked, because the MRM vastly over estimates mortality of the smallest and largest prey (Figs. 4B and 4C).

## DISCUSSION

Nonlinear processes that modify predation risk must be considered when predicting multiple predator effects. In this study we show how size-dependent risk and numerical depletion of prey interact in complex ways, which leads predictions from the MRM to be biased in ways that could be wrongfully interpreted as evidence for facilitation and inhibition among predators. Similar discrepancies can be expected with other common functional response modifiers such as temperature (*e.g.*, *Davidson et al., 2021*). However, it is difficult to evaluate to what extent failure to account for nonlinear modifiers of predation risk have affected our general understanding about the importance of emergent multiple predator effects, in part because data on these variables are often not reported.

For instance, most studies do not report size data (Supplement 2), even though numerous studies have shown that size variation is a strong determinant of predation risk (*Alford, 1999*; *Aljetlawi, Sparrevik & Leonardsson, 2004*; *McCoy & Bolker, 2008*; *McCoy et al., 2011*).

We found mismatches in the predicted survival of prey in multi-predator scenarios both when size-dependent risk declined exponentially (Fig. 2) and when prey risk was a hump shaped function of prey size (Figs. 3 and 4). In our simulations, the mismatch in the predictions between the MRM and generalized functional response was largest when there was a size refuge for large prey (small values of $d$ in Fig. 2A). When all prey were vulnerable, however, the pattern of mismatch switched directions as the effects of prey

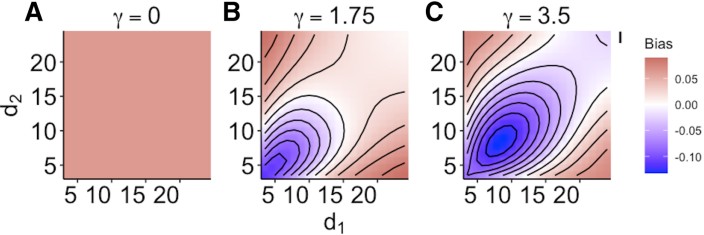

**Figure 4 (A–C) Mismatch in the predictions of the MRM and generalized functional response models when predation risk was a hump shaped function of prey size.** Each facet displays the outcome from simulated experiments that ran for 10 times steps for three different size specialization scenarios. The heatmap indicates differences between predicted survival based on the MRM and generalized functional response model. The x and y axes represent the values of the focal size parameter for the two predators Eq. (4). Negative values indicate the MRM predicted higher risk to prey (leading to conclusions of risk reduction), whereas positive values the MRM underestimated risk (leading to conclusions of risk enhancement for prey).

depletion on risk outweighed the reductions in risk associated with large size (Fig. 2A). Moreover, that magnitude of mismatch between the two models increased with experimental duration as the size structure and density of the remaining prey cohort changed through time (Fig. 2B). When prey risk is a unimodal function of size, varying both how risk scales with prey size ($d$ in Eq. (4)), and the width of the prey vulnerability window ($\gamma$ in Eq. (4)) MRM predictions can generate opposing and increasing patterns of divergence from the simulated prey survival with combined predators (*e.g.*, Figs. 3 and 4). In fact, failure to account for size dependence could lead to predicted survival of prey that was as much as 40% lower when predators specialize on small prey (Figs. 4B and 4C). However, large regions of the parameter space that we investigated showed only modest mismatches between the generalized functional response model and the MRM ( *e.g.*, Fig. 4A). This unexpected result stems in large part from the potentially opposing effects on risk of size-dependence and depletion. As prey are depleted due to predation, *per capita* risk increases through time; however, prey risk often decreases as prey grow or only larger, less vulnerable size classes remain. Said differently, size-dependent predation depletes smaller prey at a faster rate leaving a larger proportion of remaining prey in less vulnerable size classes, which decelerates prey risk in a way that counteracts the increasing risk expected with depletion. However, even when the MRM and generalized functional response predictions are consistent or even match, the MRM approach fails to capture important mechanisms leading to the observed outcomes for prey abundance. Moreover, these mismatches may become magnified with other risk modifiers such as temperature which can have asymmetrical effects on predators and prey as well as on prey of different sizes (*Davidson et al., 2021*).

The results of this study reinforce pleas for better integration of empiricism and theory in ecological studies (*Bolker et al., 2003*; *Bolker, 2008*; *Yates et al., 2018*) to help elucidate mechanistic processes that will improve our ability to predict ecological interactions (*Cottingham, Lennon & Brown, 2005*; *Denny & Benedetti-Cecchi, 2012*). We show the potential value of incorporating nonlinear modifiers of predation risk for generating predictions and for developing a better understanding of processes that underlie multiple

predator prey interactions. The increasing availability of powerful computational tools and the deployment of more efficient experimental designs (*Barraquand & Gimenez, 2021*; *Coblentz & DeLong, 2021*; *Daugaard, Petchey & Pennekamp, 2019*; *Rosenbaum & Rall, 2018*; *Uszko, Diehl & Wickman, 2020*) aimed at parameter estimation along with hypothesis testing make mechanistic studies more feasible and experimental results more informative (*Aljetlawi, Sparrevik & Leonardsson, 2004*; *Bolker, 2008*; *Daugaard, Petchey & Pennekamp, 2019*; *Denny & Benedetti-Cecchi, 2012*; *McCoy et al., 2011*; *McCoy & Bolker, 2008*; *Okuyama & Bolker, 2012*).

The promise of inferring mechanisms underlying complex ecological interactions based on deviations from linear extrapolations of simple statistical models has inspired hundreds of ecological experiments. However, failure to appreciate the important underlying mechanistic assumptions implicit in these models has obscured our general understanding of the functional effects of multiple predators in complex food webs. *McCoy, Stier & Osenberg (2012)* demonstrated how failure to account for depletion during multiple-predator experiments biased conclusions of experiments aimed at understanding the functional roles of multiple predators for regulating shared prey. Here we have extended that analysis to show that changes in prey risk due to other common nonlinear processes like body size and temperature may also need to be incorporated into predictions for multiple predator outcomes.

MPE studies often use factorial experimental designs that have high power for null hypothesis testing, and it seems likely that these experiments will continue to be commonly employed in multiple predator effects studies. Therefore, we suggest that future MPE studies need to explicitly state assumptions implicit in the experimental designs used (*e.g.*, constant *per capita* predation rates, size-independent predation, constant temperature conditions), about the sizes (mean and variances) of predators and prey, potential impacts of other strong covariates (*e.g.*, temperature), and to temper any inferences appropriately given the limitations of the statistical and experimental approaches employed. For instance, the assumptions of linear predator functional responses or zero depletion implicit with the MRM are nearly always violated; studies that find deviations from the MRM expectations should not fall back on traditional interpretations that unconditionally attribute deviations of observed prey survival from the MRM to emergent predator facilitation or inhibition. It would be more useful to emphasize the quantification of parameters that may give mechanistic insights about the processes generating observed outcomes in multiple predator-prey interactions. For example, investigators could explore different covariate effects (*e.g.*, temperature) in the generalized functional response approach presented here, or quantify how higher order modifications (*e.g.*, antipredator behaviors) of attack rates or handling times change over the course of experiments to modify interaction strengths.

Several meta-analyses and expert reviews have concluded that multiple predator—prey interactions can rarely be predicted from the independent effects of each predator alone (*Byrnes & Stachowicz, 2009*; *Griffen, 2006*; *Griffin et al., 2008*; *Schmitz, 2007*). However, since most studies have assessed the predictability of multiple predator effects using the MRM, our results demonstrate the nuance driving the failure of such generalizations. This

study highlights that our understanding of multiple predator effects may still be largely incomplete and elevates the promise of incorporating more mechanisms into multiple predator effects research for advancing the field.

Although we do not demonstrate the process here, the way to make multi-predator predictions based on the dynamics of growth and size- and density-dependent predator mortality is to parameterize nonlinear functions for growth, predator preference, and predator functional response from experimental data. Such experiments and model-fitting procedures are a core part of our research programs (*Davidson et al., 2021*; *McCoy et al., 2011*; *McCoy, Stier & Osenberg, 2012*; *McCoy & Bolker, 2008*; *Okuyama & Bolker, 2012*; *Vonesh & Bolker, 2005*). The least commonly estimated component of Eq. (1) is the diffusion term describing variation in growth rate: this term can be roughly estimated by fitting a linear regression to the time-dependent variance around an estimated growth curve, or by the more sophisticated methods described in *Brooks, McCoy & Bolker (2013)*. Granted, this procedure is more difficult than using the MRM to estimate multi-predator outcomes—which only needs a single value of survival from a series of single-predator trials—but, as we have demonstrated here, the dynamics of predator-prey systems are often much too dynamic and nuanced to be captured by the simple assumptions of the MRM.

## ACKNOWLEDGEMENTS

The authors thank the members of the Vonesh and McCoy labs for feedback on earlier drafts of this manuscript.

### Funding

This research was supported by the US National Science Foundation Grants 156686 to James R. Vonesh, 1556743 to Michael W. McCoy, and 1556729 to Jeremy Wojdak. The funders had no role in study design, data collection and analysis, decision to publish, or preparation of the manuscript.

### Grant Disclosures

The following grant information was disclosed by the authors:
US National Science Foundation: 156686, 1556743 and 1556729.

### Competing Interests

The authors declare that they have no competing interests.

### Author Contributions

- Michael W. McCoy conceived and designed the experiments, performed the experiments, analyzed the data, prepared figures and/or tables, authored or reviewed drafts of the article, and approved the final draft.
- Elizabeth Hamman performed the experiments, analyzed the data, prepared figures and/or tables, authored or reviewed drafts of the article, and approved the final draft.

- Molly Albecker performed the experiments, analyzed the data, authored or reviewed drafts of the article, and approved the final draft.
- Jeremy Wojdak conceived and designed the experiments, authored or reviewed drafts of the article, and approved the final draft.
- James R. Vonesh conceived and designed the experiments, authored or reviewed drafts of the article, and approved the final draft.
- Benjamin M. Bolker conceived and designed the experiments, performed the experiments, analyzed the data, prepared figures and/or tables, authored or reviewed drafts of the article, and approved the final draft.

### Data Availability

The code is available in the Supplemental File and at GitHub: https://github.com/eahamman/Predator.Diversity.

### Supplemental Information

Supplemental information for this article can be found online at http://dx.doi.org/10.7717/peerj.13920#supplemental-information.

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
