# Peer review of "Incorporating nonlinearity with generalized functional responses to simulate multiple predator effects"

_PeerJ, doi:10.7717/peerj.13920_

## Round 0.1 · original submission · Major Revisions

All three expert reviewers found merits in this paper, and all made detailed suggestions that I think are very useful to improve the presentation and also the numerical assessment of the model. Please address these points preparing the revised version of the paper. I am attaching through the submission server the file provided by Reviewer 3, please tell me if you do not receive it or ignore it if you receive it twice.

·

Basic reporting

In this manuscript, the authors present a novel approach to modeling multiple predator effects. They propose to model multiple predators in an ODE framework with density-dependent functional responses instead of assessing null models with a multiplication of single predation risks (MRM framework). Additionally, non-linear dependencies of functional response parameters with respect to a prey trait like body-size are incorporated, and the approach even allows to model a change of this trait distribution along time using a PDE framework.

While I think this is an interesting and original approach, I found myself asking in my first reading of the manuscript, why these two processes (multiple predation and nonlinear bodysize-dependency of interaction parameters) are modeled in conjunction, and what the individual effects of these processes would look like. Only in the discussion I picked up that this study builds on McCoy et al 2012 and adds the nonlinear bodysize-dependency, an approach partly described also in McCoy et al 2011. I propose to state this in the introduction already to motivate why these two processes are modeled in interaction and individual effects (the one or the other) are not considered here.

Experimental design

I found some formal inconsistencies that can be fixed with some simple corrections. They are listed in the additional comments below.

I’d like to encourage the authors to be a bit more descriptive of their model for the general reader. Especially the shift from an ODE on a single state to a parabolic PDE, having a distribution / range of a state variable, might be hard to grasp. A more illustrative description of the terms in Eq 1, maybe even a conceptual figure would be highly appreciated.

Validity of the findings

Clear deviances of a former framework (MRM) from the proposed method (Generalized functional response) were demonstrated in a simulation study. This study was conducted over a broad range of possible nonlinear dependencies of attack rates on prey bodysize. These findings were linked to practical advice for future MPE experiments.

Additional comments

L 85 The predation loss term is incorrect. You factored out C(s), yet C appears in the numerator of E_i (Eq. 2), also predator abundance P_i is missing from the model.

L 85 The correct notation for 2nd order derivative is (d^2 C / d s^2)

L. 85 Please be more descriptive about this equation already at this point. E.g., explain what the partial derivatives of first and second order mean / what effect they have on the distribution of C(s). This might be totally unclear for the general reader.

L 97 In Eq 3, C should read C(s).

L 102 A recent meta-analysis can be found in Uiterwaal & DeLong 2020, https://doi.org/10.1002/ecy.2975

L 119ff: For the exponential function, please use as notation either e^x or exp(x), but not exp^x (also eqns 4 & 5)

L 120. The word monotonic is used incorrectly. While eq. 3 describes a monotically decreasing function, eq 4 does not. Please use e.g. “unimodal” or “hump-shaped” instead throughout the manuscript. You could also refer to Fig 1 at this point for illustration.

L 123 “d scales the maximum attack rate” This formulation is misleading since d does not alter max. attack rate alpha, but scales s. Please rephrase.

L 133ff What is the variable T? Should be replaced with s to make it consistent with the manuscript.

L 136 “We also incorporate individual variation in growth via a simple diffusion term.” It’s not straightforward how this diffusion term models individual variation in growth. Please be more descriptive here.

L 240-241 You could refer to Fig 1a again for illustration

L 252 You could refer to Fig 1b again for illustration

L 260 I think the reference should be to Figure 4 instead of Figure 3.

L 265 Why aren’t Figs 4 b & c symmetrical along the 1-1-line? Are predators P1 and P2 different in their parametrization other than d1 and d2?

L 315-319 More recent advancements can e.g. be found in Rosenbaum & Rall 2018, Daugaard et al 2019, Uszko et al 2020, Barraquand & Gimenez 2021
https://doi.org/10.1111/2041-210X.13039
https://doi.org/10.1111/1365-2656.13053
https://doi.org/10.1002/ecs2.3051
https://doi.org/10.1016/j.tpb.2021.01.003

Figs 3 & 4. Please add panel labels a, b, c

Reviewer 2 ·

Basic reporting

No comment.

Experimental design

No comment.

Validity of the findings

No comment.

Additional comments

In their manuscript "Incorporating nonlinearity with generalized functional responses improves predictions of multiple predator effects", McCoy et al. investigate how prey depletion and nonlinearities in predation risk for prey combine to potentially bias estimates of multiple predator effects. They develop a generalized functional response model which they then parameterize in terms of prey body size variation and compare its predictions to the widely used multiplicative risk model for assessing multiple predator effects. The authors find that bias due to not accounting for prey risk changes with body size can be substantial and interacts with prey depletion. Unfortunately, the authors also show that it is difficult to assess the impact this might have on previous studies because few of them mention prey size and its variation. The authors conclude that future studies on multiple predator effects should attempt to account for prey depletion and other nonlinearities or at least state how these factors may influence their results or conclusions.

In general, I find this paper to be well-written and to be successful in its goal of evaluating how nonlinearities that alter prey risk can change conclusions in multiple predator effect studies. I do have some largely minor comments that I list below in hopes that they can help improve this already well-done manuscript.

Lines 38-42: I'm not sure if 'limited by attack rates' is exactly right. Maybe 'defined by'?

Lines 45: Could also mention the use of functional response experiments in evaluating potential biocontrol agents. For example: Cuthbert et al. 2018. Biological control agent selection under environmental change using functional responses, abundances, and fecundities; the Relative Control Potential (RCP) metric. Biological Control. 121:50-57.

Line 47: Perhaps cite Griffen, B.D. 2021. Considerations when applying the consumer functional response measured under artificial conditions. Front. Eco. Evo. doi: 10.3389/fevo.2021.713147

Line 59 and throughout: I think a better citation for nonlinear functional responses being widespread compared to the Lafferty citation might be Jeschke et al. 2004. Consumer-food systems: Why type I functional responses are exclusive to filter feeders. Biological Reviews. 79: 337-349.

Line 75: The description of the remainder of the paper only describes the next section. Perhaps you could also describe how the framework is then applied? I think this would give the reader a better roadmap of what to expect.

At least one citation, Brooks et al. 2013, is missing from the literature cited.

Lines 147-148: I think this sentence could be expanded or a sentence could be added to the beginning of the section 'Reviewing size dependence' to motivate the literature search. In the current version of the manuscript, it feels like it kind of comes out of nowhere.

Reviewing size dependence: Are the 119 papers cited somewhere? It would be nice if the authors of those papers were able to get some credit for their contribution to the field. Perhaps they could be cited in a table in the supplemental material?

Lines 231-233: This may be my biggest comment. The numbers reported here do not match those reported in the supplemental material. For example, the supplemental material says that '64% gave some indication of the size of prey included in their study.' Furthermore, this sentence says 119 studies whereas the methods say 121 studies from 119 papers.

Figures 2-4: It is unclear what 'bias' is in these figures. Would it be possible to have the color axis show a relative or percent bias or is that already what is shown in these figures? If so, it isn't clear from the figure caption.

Supplemental Material 1: There are some typos in this supplemental material.

Reviewer 3 ·

Basic reporting

In general: The article is generally of interest and shows promise. In the current write-up some information needed for a complete understanding is missing which makes the article less accessible to a broad audience. The sometimes unclear presentation also applies to the carried-out simulations and might have caused me to misunderstand part of the model. Also, the introduction of the model, the literature review and the simulations done to show the biasedness of the MRM all feel a bit disconnected to each other.

Some of my comments are listed here. Find a complete list in the attached document.

Lines 49-54: the authors should show more evidence that this method (MRM) is still being used by researchers by citing more recent cases of its use (the citations at lines 49-54 are all rather old, with the exception of the self-citation). Given that the authors already do a literature review about MPEs and prey size, they could also report how many of the found studies used the MRM.

L201-202: for text clarity, I suggest moving this sentence to an earlier place to better explain to the readers what is done and why it is done.

Citations required at lines: 24, 69-73, 278

Experimental design

General comment: The sometimes unclear presentation also applies to the model and the carried-out simulations. More importantly, the authors do not test their model on actual data, but only used it to generate data with which they show that the MRM can be biased if there is a certain structure to the data. I am not convinced that this is enough to show the usefulness and practicality of their model.

Some of my comments are listed here. Find a complete list in the attached document.

Equation 1:
- Based on the notation used, it appears that the prey consumption in a given level of s is modelled to be independent of the abundances of prey in the other levels of s. If this is correct, please justify this assumption. If this is not correct, please improve the notation used (e.g. bold characters for vectors and matrices).
- The symbols used in equation 1 and the symbols used in the text do not match. Please use a consistent mathematical notation

Simulations: the description of the simulations should be more transparent. For example:
- Based on the example code provided, it seems that the simulations were done with a prey starting abundance of 100. But as prey per capita consumption is a nonlinear function of prey abundance (and in this case of prey size), the starting population size does affect survival rates. I have not seen this mentioned in the article. It is similarly to the length of the trials (the simulations), which in Figure 2 is varied. Please do the same for the starting population and justify starting values in the main text.

The use of the model:
- By only creating simulations based on their model, the authors do not directly test their model: they do not show that their model improves the functional response estimation with real data or at least with simulated data not based on the model itself (i.e., independent data). The authors need to test their model to show that it is not just useful as a way of simulating predator-prey data with a prey-size dependency. This is currently not done. This is not the same as showing that the MRM is biased under certain conditions. I am also wondering whether there are not other more sophisticated model than the basic MRM (but not quite as complex as the presented model) to which one could compare this new model. This would make for a much more interesting comparison.

Validity of the findings

no comment

Additional comments

In the attached document I give more detailed feedback, please consider it when moving forward. In general I like the article, but some things need refinement. Best of luck.

Annotated reviews are not available for download in order to protect the identity of reviewers who chose to remain anonymous.

---

## Round 0.2 · Minor Revisions

Both reviewers found that the new version has successfully addressed their concerns. The third reviewer was not available to review the paper, but given the positive opinion of the others I do not think that this is necessary. Please submit a new version that incorporates the detailed changes suggested by Reviewer 1.

·

Basic reporting

The authors successfully revised the manuscript and considered all of my comments. There are still a couple of minor inconsistencies as addressed below. However, in my opinion, these would not require another round of reviews, and I recommend publication if these corrections are made.

L. 93: Again, please use the mathematical convention d^2 C / d s^2 for a second order partial derivative.

L. 105: Please specify if attack rate a(s) and handling time h(s) are predator-specific (a_i, h_i) or universal (a, h) in your model, also in l. 110 and 115. Later in the methods section you use identical handling times, but predator-specific attack rates, but this is not clear at this point and seeing a_i(s), but universal h(s) is confusing.

L. 111 and l. 115: It is not indicated that the sums run over index s (I assume), otherwise the reader might wrongfully imply the terms sum over index i.

L 159. Please use lowercase t on both sides of the equation.

Figure 1 legend: Please also exchange “monotonic” by “unimodel” here.

Experimental design

no comment

Validity of the findings

no comment

Reviewer 2 ·

Basic reporting

No comment.

Experimental design

No comment.

Validity of the findings

No comment.

Additional comments

I have re-read this manuscript and the authors' responses to reviewers and I have no further comments on this manuscript.

---

## Round 0.3 · accepted · Accept

The Authors have made all the minor changes suggested by the Reviewers, and the paper is ready for publication.